# SELF-MONITORING LARGE LANGUAGE MODELS FOR CLICK-THROUGH RATE PREDICTION

## ABSTRACT

Click-through rate (CTR) prediction tasks traditionally aim to model extensive user-item feature interactions. Recent approaches fine-tune Large Language Models (LLMs) using user-item features as input and click labels as output. However, due to the sparsity of click labels, the attention mechanism may focus on a subset of features rather than all features. This can hinder LLMs' ability to accurately match features to click labels, resulting in performance that does not consistently exceed traditional state-of-the-art CTR approaches. To address this, we introduce a SLLM4CTR framework which uses adaptive temperature and label matching loss to improve fine-tuning and inference process of LLMs. The adaptive temperature serves as a confidence score to calibrate CTR predictions by quantifying the LLMs' attention to user-item features. The label matching loss clearly distinguish between click-inducing and non-click-inducing features by constraining the representation space of click labels. By combining these two designs, SLLM4CTR improves feature utilization in LLMs and enhances the matching of user-item features to click labels. Experimental results demonstrate that SLLM4CTR significantly outperforms state-of-the-art baselines, including both traditional and LLM-based CTR approaches. The code will be open-sourced.

## 1 INTRODUCTION

Click-through rate (CTR) prediction uses user-item features, such as item rating, to estimate click probabilities and rank candidate items. This task is crucial for recommendations across various domains, including social media and online advertising (Zhao et al., 2022). Accurate CTR prediction requires capturing extensive feature interactions, which involves different feature combinations (e.g., item rating and price) to improve representation learning (Chen et al., 2024; Yu et al., 2021). For example, Meta's CTR prediction model heavily relies on sparse and categorical feature interactions to enhance performance (Zhang et al., 2022).

Recent approaches typically fine-tune LLMs using user and item features as the input prompt, with click labels as the target output (Geng et al., 2022; Hou et al., 2024b). These approaches take advantage of LLMs' sophisticated attention mechanism to capture complicated semantic information, which allows them to highlight certain parts of users and items (Wu et al., 2023b). To further leverage the mechanism and inform LLMs' decision-making process, recent approaches provide a broader range of features for LLMs to attend. For example, LlamaRec (Yue et al., 2023) adds estimated CTR probabilities from traditional CTR models to the prompt, while ClickPrompt (Lin et al., 2024b) includes feature representations from traditional CTR models as additional tokens to the prompt.

While the attention mechanism equips LLMs with powerful feature emphasis capabilities, it may not be sufficient on its own for CTR prediction tasks. This insufficiency arises due to the sparsity of click labels: click labels are typically absent from LLMs' pre-training corpora, and the fine-tuned click labels are usually sparse and limited. Therefore, attention may struggle with learning high-quality feature interactions and might focus on only a subset of features rather than considering all relevant ones. Consequently, LLMs may face difficulties in effectively matching user-item features with click labels, which is more severe for items with few click labels (i.e., tail items), where accurate predictions rely more heavily on intricate feature interactions. To investigate this hypothesis, we start with the following research question: **How well do fine-tuned LLMs utilize the features in click label prediction?**

To answer this question, we perform feature-wise and click-wise analyses to gain insights into the predictions made by LLMs. *Feature-wise*, we employ explainable AI techniques to visualize feature importance for a fine-tuned LLM and traditional CTR approaches. Surprisingly, we observe that the attribution score of user-item features to the final prediction is significantly lower compared to traditional CTR approaches. This aligns with our hypothesis that the LLM is making predictions with limited utilization of the user-item features (Figure 2). *Click-wise*, we compare the performances of a fine-tuned LLM with the traditional CTR approaches on head and tail items, which are items associated with more and fewer click labels, respectively, in Figure 1. While the fine-tuned LLM shows comparable performance to the best baseline on head items, its performance on tail items is notably worse, further validating our hypothesis. This suggests that LLMs may not effectively match features to click labels, as tail items rely more on feature interactions for accurate predictions. These findings indicate a potential direction for improving LLM in CTR tasks: **enhancing the LLMs' ability to effectively use all relevant features and better match user-item features to labels.**

It is challenging to advance in this direction primarily due to two reasons. *(i)* Inconsistent feature attention across different predictions. Pre-trained on general text, LLMs often prioritize features inconsistently across different predictions. Their attention mechanism might focus on different features for similar inputs, resulting in varied feature attention. *(ii)* The power law distribution of CTR datasets. There is an imbalance in the distribution of item occurrences in the dataset. The tail items are infrequently present in training samples. Some feature interactions associated with click labels for these tail items are rarely observed. Consequently, fine-tuning LLMs to match these rare features to click labels is difficult due to the limited training samples available.

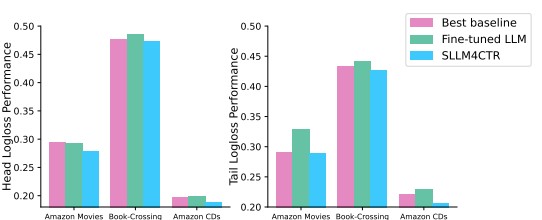

Figure 1: Logloss comparison on the head (left) and tail (right) items among the best traditional CTR baseline, simply fine-tuned LLM, and the proposed SLLM4CTR. The smaller, the better.

To address these challenges, we propose Self-monitoring LLMs for CTR (SLLM4CTR) which uses two simple yet effective designs to improve the fine-tuning and inference process of LLMs. To enhance user-item feature utilization, we introduce an adaptive temperature, a confidence score that directly calibrates CTR predictions. This temperature associates LLM attention to user-item features with estimated CTR probability, encouraging LLMs to correlate predictions with features in the representation space. To improve the matching of user-item features to click labels, we introduce a label matching loss, which constrains the representation space of click labels. This loss promotes compact representations for click and non-click labels, enabling LLMs to clearly distinguish between click-inducing and non-click-inducing features, even for items with limited click labels. By combining these two designs, SLLM4CTR enhances LLMs' attention to user-item features and better matches user-item features to click labels. Extensive experiments demonstrate that SLLM4CTR significantly outperforms state-of-the-art traditional and LLM-based CTR approaches. In summary, we make the following contributions:

- We conduct feature-wise and click-wise analyses to understand how well a fine-tuned LLM utilizes features in click label prediction. Our findings reveal that simply fine-tuned LLMs exhibit limited utilization of user-item features and do not match user-item features and click labels well.

- We introduce SLLM4CTR, which incorporates two simple yet effective designs to address the problem. Feature-wise, we propose an adaptive temperature to associate LLMs' attention on user-item features to their predictions. Click-wise, we introduce a label matching loss to differentiate between click-inducing and non-click-inducing features.

- Experimental results on three real-world datasets show that SLLM4CTR significantly outperforms state-of-the-art baselines. Further analysis validates that SLLM4CTR can enhance the utilization of user-item features and improve the matching of user-item features to click labels.

## 2 PRELIMINARIES

### 2.1 PROBLEM STATEMENT

We denote a CTR dataset $\mathcal{D}$ as $(\mathcal{U}, \mathcal{I}, \mathbf{R})$. $\mathcal{U} = \{\mathcal{U}_1, \mathcal{U}_2, ..., \mathcal{U}_U\}$ represents $U$ user profiles, where each user profile $\mathcal{U}_u$ contains a list of textual and non-textual features, such as age and location. Similarly, $\mathcal{I} = \{\mathcal{I}_1, \mathcal{I}_2, ..., \mathcal{I}_I\}$ denotes $I$ item profiles, where each item profile $\mathcal{I}_i$ includes features like title and description. $\mathbf{R} \in \mathbb{R}^{U \times I}$ is the matrix, where each value $r_{ui} \in \mathbf{R}$ gives the label justifying whether the user $u$ clicks the item $i$ or not. To leverage LLMs for CTR prediction, we verbalize the template function with $\mathcal{U}_u$ and $\mathcal{I}_i$ and get a list of prompt tokens $\{x_1, x_2, \ldots, x_L\}$ of length $L$, where $x_{click}$ is the next click label token that needs to be predicted by LLM, i.e., "Yes" or "No". We input the prompt tokens into LLM and get their hidden representation $\{\mathbf{e}_1, \mathbf{e}_2, \ldots, \mathbf{e}_L\}$ at the last transformer layer. And the two weight vectors $\mathbf{w}_{Yes}$ and $\mathbf{w}_{No}$ in the LLM's head layer will project $\mathbf{e}_L$ to click and non-click scores $l(x_{Yes}|x_1, x_2, \ldots, x_L)$ and $l(x_{No}|x_1, x_2, \ldots, x_L)$. Our objective is to maximize the probability of predicting the correct click label token $p(x_{click})$, where probabilities $p(x_{Yes})$ and $p(x_{No})$ are obtained by applying the softmax function to $l_{Yes}$ and $l_{No}$.

### 2.2 PROMPTING AND FINE-TUNING STRATEGIES

**Prompt Templates.** We formulate a prompt template for each dataset to encode user and item features. Additionally, we include user ID and item ID, as the previous research has shown their effectiveness in personalization (Yuan et al., 2023). For the Amazon Movies dataset, we employ the following prompt template (templates for other datasets are detailed in the Appendix A.4):

*Given the user's and item's attributes, identify whether the user will like the target movie by answering Yes or No. Here is the information of the user [user_id]. Here is the information of the movie [movie_id]: Its title is [title], and its price is [price]. Its description is: [description]. Its sales rank among all movies is [sales rank].Response:[ ]*

We fill the blank $[\cdot]$ with features $\mathcal{U}_u, \mathcal{I}_i$ to obtain the prompt text. The LLM is expected to output "Yes" or "No" for the last empty token, which corresponds to $r_{ui} = 1$ or $r_{ui} = 0$, respectively following previous effort (Geng et al., 2022). To optimize the target token predicted probability, we employ parameter-efficient fine-tuning (PEFT) LoRA (Hu et al., 2022) for fine-tuning LLMs.

## 3 ANALYSIS OF SIMPLY FINE-TUNED LLMS

In this section, we conduct feature-wise and click-wise analysis to gain insights into the predictions made by LLMs by answering: (i) how much do the features contribute to the label prediction (Section 3.1)? (ii) how do fine-tuned LLMs match user-item features with click labels (Section 3.2)?

### 3.1 FEATURE ATTRIBUTION SCORE ANALYSIS

To answer the question (i), we quantitatively attribute the label prediction to its input features. To achieve this, we employ integrated gradient attributions (Sundararajan et al., 2017). Since features are a list of tokens, we attribute the predictions to input tokens. The contributions of input tokens to the predictions $p(x_{click})$ are defined as the gradients of the LLM input token embedding. Let $g(\cdot)$ denote the attribution function and boldcase letter $\mathbf{x}_i$ denote the input token embedding of $x_i$. The attribution score for a input token embedding $\mathbf{x}_i$ is:

$$g(\mathbf{x}_i) = (\mathbf{x}_i - \mathbf{x}_{base}) \cdot \sum_{q=1}^{m} \frac{\partial p_{x_{click}}(\mathbf{x}_{\text{base}} + \frac{q}{m}(\mathbf{x}_q - \mathbf{x}_{\text{base}}))}{\partial \mathbf{x}_q} \cdot \frac{1}{m}, \tag{1}$$

where $\mathbf{x}_{base}$ represents the starting point of the integration, which is usually chosen to be a zero vector. Starting from $\mathbf{x}_{base}$, we construct a linear interpolation path towards the input token embedding $\mathbf{x}_i$ using $m$ uniform steps. At the $q_{th}$ step, we compute the gradient of target label prediction $p(x_{click})$ w.r.t to the input $\mathbf{x}_q$, where $\mathbf{x}_q$ is obtained through linear interpolation between $\mathbf{x}_{base}$ and $\mathbf{x}_i$.

Since gradients naturally measure how changes in input features affect the model's output, the equation 1 accumulates gradients along the interpolation path from a starting point to the input. This

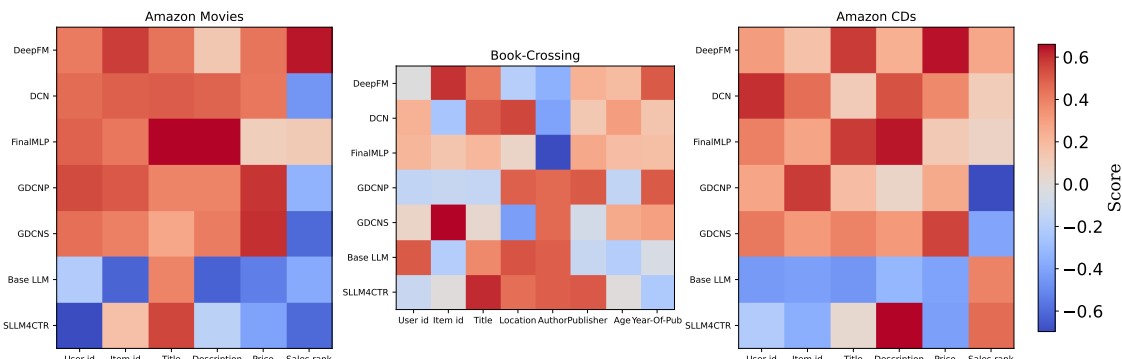

Figure 2: Attribution scores for Amazon Movies, Book-Crossing, and Amazon CDs datasets. The first five models are traditional CTR models that explicitly learn extensive feature interactions. The second last and last models are the simply fine-tuned LLM and SLLM4CTR. Simply fined-tuned LLM exhibits only 1, 4, and 1 features with positive attribution scores across the three datasets, while the proposed SLLM4CTR addresses this issue with 2, 4, and 3 features that contribute positively.

Table 1: Examination of learned feature-click matching based on head/tail item comparison between traditional CTR baselines, simply fine-tuned LLMs (Base), and SLLM4CTR (Ours).

| Model | Amazon Movies | | Book-Crossing | | Amazon CDs | |
|---|---|---|---|---|---|---|
| | Head Logloss↓ | Tail Logloss↓ | Head Logloss↓ | Tail Logloss↓ | Head Logloss↓ | Tail Logloss↓ |
| DeepFM | 0.2941 | 0.3001 | 0.4813 | 0.4336 | 0.2030 | 0.2212 |
| DCN | 0.2943 | 0.2925 | 0.4802 | 0.4326 | 0.1964 | 0.2239 |
| FinalMLP | 0.3013 | 0.2962 | 0.4769 | 0.4336 | 0.2124 | 0.2216 |
| GDCNP | 0.2973 | 0.3006 | 0.4780 | 0.4359 | 0.2026 | 0.2218 |
| GDCNS | 0.2937 | 0.2906 | 0.4828 | 0.4379 | 0.2024 | 0.2210 |
| Base | 0.2932 | 0.3277 | 0.4850 | 0.4421 | 0.1994 | 0.2295 |
| SLLM4CTR | **0.2786** | **0.2884** | **0.4722** | **0.4266** | **0.1886** | **0.2056** |

path integral captures how each input token progressively contributes to transforming the model's prediction from the prediction of starting point to the prediction of input proved by the theorem:

**Theorem 1.** *The sum of all input token attribution scores $\sum_{i=1}^{L} g(\mathbf{x}_i)$ exactly equals the difference in click label prediction probabilities between $p(x_{click}|\{x_i\}_{i=1}^{L})$ and $p(x_{click}|\{x_{base}\}_{i=1}^{L})$.*

The principle behind this theorem is that integrated gradients decompose the target label prediction probabilities into a weighted sum of input tokens gradient integrals computed over a specified interval. And we provide its proof in Appendix A.1. Along this line, a positive attribution score for a feature indicates that it helps LLMs predict the target labels more accurately. Increasing the number of features with positive attribution scores can enhance related feature interactions, enabling LLMs to better recognize target labels. Using equation 1, we can get get the attribution score for any token in one prompt of testing set and more details are in Appendix A.6. Then, we sum all token scores within the feature across all samples in the testing set to obtain the feature score. Finally, for each feature, we add up the score from each instance in the test set and get the overall feature score. Notably, traditional CTR models have one embedding of the feature instead of multiple token embeddings. We apply the attribution score analysis similarly to the traditional CTR models.

**Attribution Score Visualization.** We simply fine-tune LLaMA-7B with the strategy introduced in subsection 2.2 on three CTR datasets: Amazon Movies, Book-Crossing, and Amazon CDs. Additionally, we train several representative traditional CTR models for comparison following the protocol described in subsection 5.1. Figure 2 illustrates their attribution scores. We observe that **the features do not significantly positively contribute to the prediction of the simply fine-tuned LLMs.** In most cases, features do not exhibit high importance in LLM's prediction. For example, in the Amazon movies and CDs dataset, only item title and sales rank significantly contribute to the prediction. In contrast, traditional CTR models assign high importance to almost all features in the prediction of these two datasets. Moreover, the overall feature scores of traditional CTR models

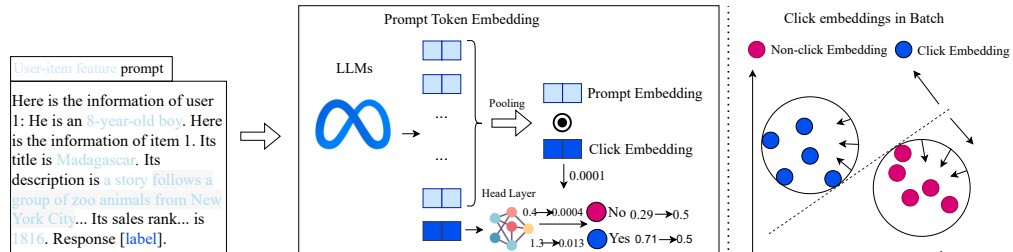

Figure 3: The pipeline of SLLM4CTR framework. In the left part, we describe the input prompt associated with the user-item features. In the middle part, we utilize the prompt and click embedding to adjust the predicted click probability. In the right part, we compact the representation space and cluster samples with the same click label in the batch.

remain higher than those of fine-tuned LLMs. These visualizations suggest the effectiveness of traditional CTR models that explicitly craft extensive feature interactions, which could be utilized to learn the complicated relationships between features and clicks based on CTR data. Conversely, LLMs are initially designed for understanding complicated textual data and generating new text. They may not easily learn the relationship well by fine-tuning simply. Therefore LLMs could get performance improvement by better utilizing features in the prediction.

## 3.2 PERFORMANCE COMPARISON ON HEAD/TAIL ITEMS

To answer the question (ii), we categorize items into head and tail groups, focusing our analysis on the model's performance for tail items. This focus is due to tail items having insufficient click samples for learning click predictions, thus relying more heavily on effective feature-to-click label matching for accurate predictions. Following previous work (Chen et al., 2022; Zhou et al., 2023), the top 20% of items with the most samples are grouped as head items while the remaining are grouped as tail items. We calculate the Logloss on each item group respectively in Table 1. We observe that **the performance on tail items in simply fine-tuned LLMs is worse than the traditional CTR models while the performance on head items in simply fine-tuned LLMs is comparable with traditional CTR models.** One possible explanation is that for head items, LLMs can leverage the abundance of samples to accurately predict the click label through fine-tuning. For infrequently appeared items in the training set, i.e., tail items, LLMs may not easily predict the labels. But since LLMs effectively manipulate text, if LLMs can match user-item textual features with click labels well, the performance on tail items will be satisfactory. However, due to the unsatisfactory performance on tail items, LLMs may struggle to match the user-item features with click labels.

## 4 SLLM4CTR

In this section, we present SLLM4CTR which includes two plug-and-play designs, as illustrated in Figure 3. Feature-wise, an adaptive temperature provides high confidence for LLMs' predictions that are highly attentive to user-item features and penalizes low-confidence predictions with large training loss (Section 4.1). Click-wise, a label matching loss is introduced to compact the representation space of click and non-click labels to assist LLMs in differentiating between click-inducing and non-click-inducing features (Section 4.2).

### 4.1 ADAPTIVE TEMPERATURE

The key idea is to leverage the correlation between the feature and click embeddings as the confidence score to dynamically refine the click probability $p_{Yes}$ and click prediction loss $\mathcal{L}_{click}$, This refinement guides LLMs to effectively associate predictions with features. The embeddings of prompt tokens contain rich contextualized feature information and serve as indicators of feature modeling. When the click embeddings of training samples are not correlated with the prompt token embeddings, the LLMs may fail to fully capture the feature-click relationships and are assigned a low confidence. To address this, we introduce a self-monitoring mechanism that uses the correlation between these embeddings

as learnable temperatures to calibrate the predicted click probability. This temperature modulates the behavior of the LLM during both training and testing phases. During training, it increases the click prediction loss, resulting in smaller gradients for low-confidence predictions. This has two effects during optimization: (i) the small gradients have a limited impact on model parameter updates, and (ii) to reduce the training loss, the LLM increases feature utilization for this sample. Together, these two effects cause the LLM to focus on this sample with enhanced feature utilization. During testing, the temperature results in a lower click probability for less correlated samples, thereby improving the ranking order of items with better feature modeling. This approach ensures that the LLM more effectively aligns predictions with feature information. To calculate the temperature, we first apply meaning pooling to the learned prompt token embeddings of the specific training sample and get the prompt embedding $\mathbf{e}_c$, defined as $\mathbf{e}_c = \frac{1}{L} \sum_{i=1}^{L} \mathbf{e}_i$. And we use the cosine similarity function $s(\cdot)$ to obtain the initial temperature. To prevent learning trivial correlation, we also try to discriminate the temperature value with the value obtained from click embedding $\mathbf{e}_{click}$ and other prompt embedding $\mathbf{e}'_c$ in the batch $\mathbb{B}$. Here we do not remove those template token embeddings from prompt embedding calculation to increase the discrimination difficulty. Since these template tokens are the same across all samples, these tokens encourage the LLMs to focus more on discriminated user-item features. And the final adaptive temperature $T$ of one sample is denoted as:

$$T = \frac{\exp(s(e_c, e_L))}{\sum_{(e'_c, e_L) \in \mathbb{B}} \exp(s(e'_c, e_L))}, \tag{2}$$

where $\exp(\cdot)$ represents the exponential function. Next, we multiply the learned temperature $T$ with the click score in the softmax function: $p_{Yes} = \frac{\exp(l_{Yes} \cdot T)}{\exp(l_{Yes} \cdot T) + \exp(l_{No} \cdot T)}$, where $l_{Yes}$ is the click score predicted by LLMs. Since we multiply the temperature instead of dividing it, a small temperature is unlikely to result in NAN values. Then we utilize the calibrated probability in the click prediction loss, i.e., cross-entropy loss $\mathcal{L}_{click}$:

$$\mathcal{L}_{click} = -(y \log(p_{Yes}) + (1 - y) \log(1 - p_{Yes})), \tag{3}$$

where $y = 1$ when $x_{click} = $ "Yes" and $y = 0$ when $x_{click} = $ "No". We then present the following theorem to provide the rationale for our design and we provide its proof in Appendix A.2:

**Theorem 2.** *The gradient of the LLMs click score satisfies the following:* $\frac{\partial \mathcal{L}_{click}}{\partial l_i} \propto T$.

The theorem demonstrates that the gradient of the click prediction loss is controlled by the temperature. When the correlation between prompt embeddings and click embeddings is low, the temperature decreases, resulting in a smaller gradient. This encourages LLMs to focus on more features in the click prediction task, effectively penalizing narrow feature attention.

## 4.2 LABEL MATCHING LOSS

To better match user-item features and click labels, especially for items with limited click labels, we introduce a label matching loss to make the boundary between click and non-click labels more distinguishable in the representation space.

To solve the issue, we introduce a training objective termed label matching loss. It learns a compact representation space for click embeddings of both click and non-click labels. By constraining the click embeddings of tail items with other samples of the same click label within the batch, LLMs can establish a clearer click decision boundary and better match user-item features and clicks.

To compact the click embeddings in the representation space, we first get the expression for the volume of the representation space occupied by click embeddings via the following theorem and then compact this volume. The proof of the theorem is provided in Appendix A.3:

**Theorem 3.** *Assume the rows in click embedding matrix* $\mathbf{E}$ *have zero mean values. The volume of the space Vol($\mathbf{E}$) satisfies the following:* $Vol(\mathbf{E}) \propto \log \det \left( \mathbf{I} + \frac{d}{|\mathbb{B}|\alpha^2} \mathbf{E}^\top \mathbf{E} \right)$,

where the batch of click embeddings represented by $\mathbf{E} \in \mathbb{R}^{|\mathbb{B}| \times d}$; $|\mathbb{B}|$ is the batch size and $d$ is the embedding dimension; $\det(\cdot)$ denotes the determinant function; $\mathbf{I}$ represents the identity matrix; and $\alpha$ denotes as a hyper-parameter.

Table 2: Comparison with traditional CTR baselines where $p$-value $< 0.01$. Notably, 0.001 AUC performance improvement can be seen as a significant performance improvement (Mao et al., 2023).

| Datasets | Metrics | Traditional CTR Baselines | | | | | Ours |
|---|---|---|---|---|---|---|---|
| | | DeepFM | DCN | FinalMLP | GDCNP | GDCNS | SLLM4CTR |
| Amazon Movies | AUC↑ | 0.8567 | 0.8559 | 0.8547 | 0.8581 | 0.8580 | **0.8710** |
| | Logloss↓ | 0.2969 | 0.2938 | 0.2991 | 0.2987 | 0.2929 | **0.2823** |
| Book-Crossing | AUC↑ | 0.7528 | 0.7533 | 0.7543 | 0.7513 | 0.7530 | **0.7791** |
| | Logloss↓ | 0.4514 | 0.4504 | 0.4497 | 0.4516 | 0.4546 | **0.4436** |
| Amazon CDs | AUC↑ | 0.8865 | 0.8882 | 0.8827 | 0.8883 | 0.8867 | **0.9053** |
| | Logloss↓ | 0.2136 | 0.2123 | 0.2178 | 0.2139 | 0.2133 | **0.1985** |

Next, we calculate the volume of spaces spanned by click and non-click labels separately using the above expression. For the volume of the space spanned by click labels, we use the click probabilities $p_{yes}$ to weigh the click embeddings in the batch. Low probability makes click embeddings contribute less to the volume. And we conduct similar operations for the space spanned by the non-click labels. The sum of the two volumes is defined as $\sum_{k=0}^{1} \frac{\text{tr}(\mathbf{\Pi}_k)}{2|\mathbb{B}|} \cdot \log \det \left( \mathbf{I} + \frac{d}{\text{tr}(\mathbf{\Pi}_k)\alpha^2} \mathbf{E}^\top \mathbf{\Pi}_k \mathbf{E} \right)$ where $\mathbf{\Pi}_k$ is a diagonal matrix, where the $i$-th diagonal element represents the probability of the $i$-th sample being associated with the non-click label when $k = 0$ and the click label when $k = 1$. By compressing the volume of these spaces, LLMs could learn a more clear click decision boundary. But it may also take the risk of collapsing the representation space gradually to a single point. To prevent this collapse and maintain meaningful representations, we also widen the space between click and non-click labels in the batch (Yu et al., 2020). The label matching loss $L_{mat}$ is formally denoted as:

$$L_{mat} = \sum_{k=0}^{1} \frac{\text{tr}(\mathbf{\Pi}_k)}{2|\mathbb{B}|} \cdot \log \det \left( \mathbf{I} + \frac{d}{\text{tr}(\mathbf{\Pi}_k)\alpha^2} \mathbf{E}^\top \mathbf{\Pi}_k \mathbf{E} \right) - \frac{1}{2} \log \det \left( \mathbf{I} + \frac{d}{|\mathbb{B}|\alpha^2} \mathbf{E}^\top \mathbf{E} \right). \quad (4)$$

The above loss has two merits: (i) Theoretical guarantee: Theorem 3 proves that optimizing the label matching loss is equivalent to compacting the representation space. (ii) No need for additional sample construction: CTR datasets suffer from label imbalance (Muhamed et al., 2021), where one label may have a large number of samples while another has a few. Directly pushing click embeddings belonging to the same label closer together to compact the representation space can be hindered by this label sparsity issue. However, manually constructing more samples for the tail class may require multiple forward propagations, which is inefficient. We combine the click prediction loss and label matching loss to fine-tune LLMs. The overall training loss $\mathcal{L}$ is defined as:

$$\mathcal{L} = \mathcal{L}_{click} + \beta \mathcal{L}_{mat}, \quad (5)$$

where the hyper-parameter $\beta$ controls the magnitude of the label matching loss.

## 5 EXPERIMENTS

Our experiments aim to answer the following research questions: **RQ1:** How does SLLM4CTR perform compared with the state-of-the-art LLM-based CTR baselines and traditional CTR baselines? **RQ2:** How does SLLM4CTR utilize the features in click label prediction? **RQ3:** How does each design component contribute to SLLM4CTR's performance improvement, and what is the associated time cost? **RQ4:** How well does SLLM4CTR generalize across different scenarios, including various backbones, and LLM-based CTR backbones? **RQ5:** What is the impact of different hyper-parameters?

### 5.1 EXPERIMENTAL SETTING

**Data Preparation.** We conduct our experiments on recommendation datasets with raw texts available. Specifically, we utilize three real-world datasets: Amazon CDs (Hou et al., 2024a), Book-Crossing (Ziegler et al., 2005), and Amazon Movies (Hou et al., 2024a). Following previous efforts (Bao et al., 2023b), we map user ratings into click labels. In the Book-Crossing dataset, we assign the true label, i.e., "Yes" to ratings greater than 5, and assign the false label, i.e., "No" with ratings less than 5. In Amazon datasets, the threshold is 3. Lastly, we also follow the previous

Table 3: Comparison with LLM-based CTR predictors where $p$-value $< 0.01$.

| Datasets | Metrics | LLM-based Baselines | | | | | Ours |
|---|---|---|---|---|---|---|---|
| | | BAIU | ClickPrompt | TallRec | P5 | LlamaRec | SLLM4CTR |
| Amazon Movies | AUC↑ | 0.8560 | 0.8623 | 0.8559 | 0.8465 | 0.8569 | **0.8710** |
| | Logloss↓ | 0.2913 | 0.2889 | 0.2962 | 0.3080 | 0.2975 | **0.2823** |
| Book-Crossing | AUC↑ | 0.7527 | 0.7561 | 0.7532 | 0.7492 | 0.7364 | **0.7791** |
| | Logloss↓ | 0.4509 | 0.4507 | 0.4529 | 0.4581 | 0.4954 | **0.4436** |
| Amazon CDs | AUC↑ | 0.8832 | 0.8919 | 0.8862 | 0.8858 | 0.8850 | **0.9053** |
| | Logloss↓ | 0.2166 | 0.2062 | 0.2385 | 0.2167 | 0.2238 | **0.1985** |

practice (Wang et al., 2023b) and randomly partition the datasets to training, validation, and test set by the ratio 80%:10%:10%. The detailed dataset statistics are shown in Table 4 in the Appendix.

**Evaluation Metrics.** Following prior work (Tian et al., 2023), we evaluate all model performance through two widely used metrics: AUC (Area Under Curve) and Logloss (Logistic loss). The AUC metric quantifies the ability of the model to distinguish positive and negative instances. The Logloss metric measures the loss caused by the difference between the predicted probability and the labels.

**Baselines.** These incorporated representative baselines for comparison can be categorized into two groups: (i) traditional CTR baselines: they are all traditional recommendation models that map the tabular features into one-hot vectors and model high-order feature interactions including **Deepfm** (Guo et al., 2017), **DCN** (Wang et al., 2021), **GDCNS** (Wang et al., 2023b), **GDCNP** (Wang et al., 2023b), **FinalMLP** (Mao et al., 2023), etc. (ii) LLM predictors: we use the same LLM as the backbone including **BAIR** (Hou et al., 2024a), **TallRec** (Bao et al., 2023b), **P5** (Geng et al., 2022), **LlamaRec** (Yue et al., 2023), etc. (iii) LLM+CTR predictors: we use the same LLM and DeepFM as the backbone including **BAIU** (Yang et al., 2023a), **ClickPrompt** (Lin et al., 2024b) **KAC** (Xi et al., 2024) **FLIP** (Wang et al., 2024a) The detailed introduction is put in Appendix A.5. We incorporate the following variant of SLLM4CTR: **Base**, **w/o Label Matching Loss**, **w/o Adaptive Temperature**, **w/o Label Matching Loss-1** and **w/o Label Matching Loss-2**.

**Implementation details.** We use the publicly released codes for all baselines. For all traditional baselines, we adopt the DeepCTR framework (Shen, 2017) for feature processing. All models are tuned to be optimal based on the validation set. We select the publicly available LLaMA-7B (Touvron et al., 2023) as the backbone for LLM-based CTR predictors in the main comparison. And we try other LLM LLaMA-2-7B (Touvron et al., 2023), and LLaMA-3-8B (AI@Meta, 2024) as the backbone in the generalization experiment. For other settings, please refer to the Appendix A.6

## 5.2 MAIN COMPARISONS (RQ1 & RQ2)

We compare SLLM4CTR with the two groups of baselines on three real-world datasets in Table 2, Table 3 and Appendix Table 5. We have several observations. ① **Modeling complicated feature interactions is helpful in CTR prediction.** In the first group, the baseline DeepFM generally performs worse than the other traditional CTR baselines which design a more effective crossing network to capture the high-order feature interactions. As the network layers deepen, these networks could implicitly model complex feature interactions. The enhancement demonstrates their capability to identify effective feature interactions. ② **Adding more features in the prompt can further boost the performance of LLM-based CTR approaches.** The P5 fine-tunes LLM with the click label simply and does not achieve satisfactory performance. The LLM + CTR models in the second group, e.g., ClickPrompt or BAIU, achieves better performance over the other. We infer that their advantages are: (i) LLMs provide semantic information that traditional CTR models struggle to capture, and (ii) traditional CTR models can better match features with click labels and provide more features for LLMs. These two models complement each other's strengths, leading to improved performance. However, they still perform far behind our models, which demonstrates that LLM + CTR may not fully stimulate the potential of LLMs' power in the CTR task. ③ **SLLM4CTR enhances the LLMs' utilization of user-item features and better match user-item features to click labels.** From the results on Figure 2 and Figure 8, our framework utilizes more features that positively contribute to target label prediction given the adaptive temperature and addresses the first issue; As shown in Figure 5, the click embedding visualization shows that SLLM4CTR learns a more

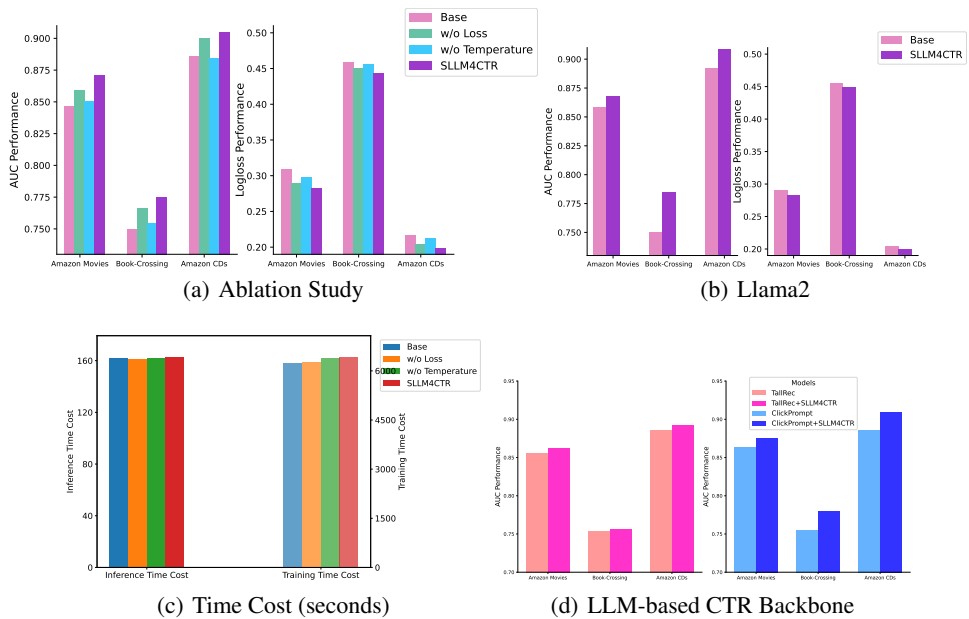

(a) Ablation Study
(b) Llama2
(c) Time Cost (seconds)
(d) LLM-based CTR Backbone

Figure 4: Figure (a) shows the performance contribution of each design, while (c) illustrates their per-epoch time costs on the Book-Crossing dataset. Figure (b) compares SLLM4CTR and its backbone on Llama2. Figure (d) does the same for LLM-based CTR backbones TallRec and ClickPrompt

compact representation space for click and non-click labels respectively. Besides, as shown in Table 1, SLLM4CTR significantly enhances performance on tail items. Both results jointly show that SLLM4CTR improves the matching features with click labels. Since tail items appear infrequently in the training set, they rely more on LLMs to effectively match user-item features with click labels. The improved feature utilization enables more features for predicting target labels of tail items. Additionally, better feature matching with click labels further enhances the performance for tail items.

## 5.3 ABLATION STUDY, EFFICIENCY & GENERALIZATION (RQ3 & RQ4)

We perform an ablation study by analyzing each design individually to understand their contributions and the associated time cost. In addition, to examine the generalizability SLLM4CTR, we apply our framework on (i) different LLM backbones and (ii) different LLM-based CTR backbones. We have the following observations based on the results in Figure 4 and Figure 7. ④ **Each design component in SLLM4CTR enhances model performance. And the added computational overhead remains minimal even expanding the dataset by 20 times.** The base model's performance is generally unsatisfactory. Two main issues are identified: inconsistent feature attention, and unclear click decision boundaries due to rarely appearing features in prompts. Both variants (without adaptive temperature and without label matching loss) show improvement over the base model, with the latter performing better. We infer that without temperature adjustment, the model may not prioritize features in click embeddings effectively. Notably, combining both designs leads to significant performance enhancement, suggesting their complementary nature. Our framework introduces no additional learnable parameters and relies solely on click embeddings, ensuring only marginal computational overhead compared to the base model. The well-trained SLLM4CTR serves as a data augmentor. It simulates realistic user behaviors offline. This simulation generates new potential user interactions. Such an approach avoids the high costs of direct online deployment. ⑤ **SLLM4CTR enhances the performance of the other LLM backbones and LLM-based CTR backbones.** We observe that the SLLM4CTR enhances the performance on Llama-2-7B and Llama-3-8B. This demonstrates that LLMs may struggle with feature utilization due to the sparsity of click labels even for the advanced LLMs. Our framework guides these LLMs to focus on informative user-item features, resulting in improved performance. Besides, SLLM4CTR unlock greater potential within other LLMs-based CTR predictors.

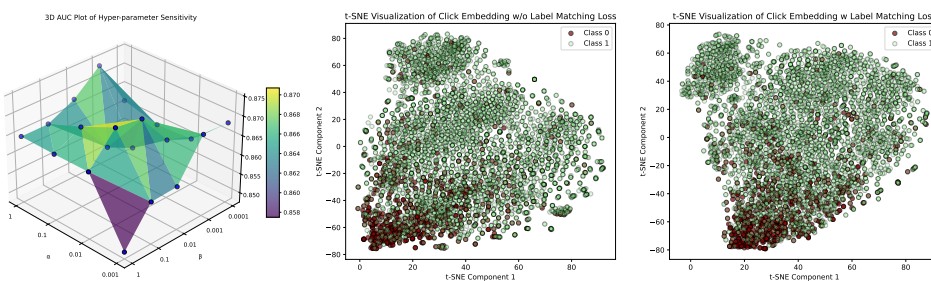

(a) Hyper parameter Sensitivity   (b) Click Embedding Visualization w/o & w/ label matching loss

Figure 5: Figure (a) shows hyper parameter sensitivity of SLLM4CTR w.r.t $\alpha$, $\beta$ on Amazon Movies dataset, presented as a three-dimensional plot using the Logloss metric. We have omitted the performance value when $\alpha = 0.0001$ since the model gets crashed in training. Figure (b) shows the t-SNE click embedding visualization w/o and w/ label matching loss of SLLM4CTR.

## 5.4 HYPERPARAMETER SENSITIVITY (RQ5)

To test the sensitivity of the hyper-parameters $\alpha$, $\beta$, we vary one hyper-parameter value and fix the other. The results on the Amazon Movies dataset are shown in Figure 5(a) and Figure 9 while the results on the other two datasets are shown in Figure 8 in Appendix. From the results, we observe: ⑥ **Generally, intermediate values of $\alpha$ and $\beta$ tend to lead to better results.** For hyper-parameter $\alpha$, larger values expand the learned representation space for click and non-click labels. However, if too large, LLMs may struggle to differentiate click labels, significantly degrading performance. Conversely, small $\alpha$ values could compress all click labels to a single point in the representation space, severely impacting model performance. For hyper-parameter $\beta$, small values limit the benefits of label matching loss in strengthening feature-label matching. Conversely, large $\beta$ values may interfere with normal click prediction, leading to performance degradation.

**Related Work: Pre-trained Language Models for CTR.** Text embeddings from pre-trained language models have been explored to enhance traditional recommender systems (Lin et al., 2024a). Recent advancements involve utilizing valuable semantic information from pre-trained language models like BERT for exploration purposes (Wang et al., 2023a; Yang et al., 2023b). For example, Microsoft Bing Ads uses the embedding of TwinBERT to facilitate precise advertising Lu et al. (2020). Furthermore, BST and BERT4CTR combine the outputs of language models and non-textual features through one additional layer Chen et al. (2019); Wang et al. (2023a). Instead of using text embeddings, we show that LLMs' inherent reasoning abilities can conduct CTR prediction directly.

## 6 CONCLUSION AND FUTURE WORK

This study investigates how fine-tuned LLMs utilize user-item features and match features with click labels in CTR tasks. Our feature-wise and click-wise analyses reveal that purely fine-tuned LLMs inadequately attend to user-item features and struggle with feature-to-click matching. To address these limitations, we introduce SLLM4CTR, which incorporates two simple yet effective designs to enhance LLM fine-tuning and inference. The first, adaptive temperature, calibrates click probability based on LLM attention to user-item features, increasing the click prediction loss when attention is insufficient. The second, label matching loss, compacts representations of click and non-click labels, helping LLMs distinguish click-inducing and non-click-inducing features for click prediction. These designs jointly promote attention towards informative user-item features and feature-to-click matching, resulting in significant performance improvements over state-of-the-art traditional CTR and LLM-based CTR approaches. Future work will explore how different prompt strategies and temperature calibration methods affect LLM performance.

REPRODUCIBILITY STATEMENT

All experiments of this paper were conducted using publicly available datasets, which are clearly cited in our paper. The code for our models and experiments will be made available in a public GitHub repository upon acceptance of the paper. This repository will include detailed documentation, requirements file for environment setup, and scripts to reproduce our experiments.

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

# A APPENDIX

## A.1 PROOF OF THEOREM 1

**Proof**: To simplify the notation, we represent $p(x_{click}|\{x_i\}_{i=1}^{L})$ and $p(x_{click}|\{x_{base}\}_{i=1}^{L})$ as $P(x)$ and $P(\bar{x})$ respectively. Let $\gamma(\alpha), \alpha \in [0, 1]$ represent a parametric curve connecting $x$ and $\bar{x}$, where $\gamma(0) = \bar{x}, \gamma(1) = x$, then we have

$$P(x) - P(\bar{x}) = P(\gamma(1)) - P(\gamma(0)) \tag{6}$$

$$= \int_0^1 \frac{dP(\gamma(\alpha))}{d\alpha} d\alpha \tag{7}$$

$$= \int_0^1 \langle \nabla_\gamma P(\gamma(\alpha)), \gamma'(\alpha) \rangle d\alpha \tag{8}$$

$$= \sum_i^L \int_0^1 [\nabla_\gamma P(\gamma(\alpha))]_i [\gamma'(\alpha)]_i d\alpha \tag{9}$$

And we set $\gamma(\alpha)$ as the straight line between two points, namely:

$$\gamma(\alpha) = (1 - \alpha)x + \alpha\bar{x} \tag{10}$$

Then the different of predictions becomes:

$$P(x) - P(\bar{x}) = \sum_i^L \left| \int_0^1 \nabla_\gamma P(\gamma(\alpha))|_{\gamma(\alpha)=(1-\alpha)x+\alpha\bar{x}} \, d\alpha \right|_i |\bar{x} - x|_i \tag{11}$$

$$= \sum_i^L \left| \left[ \frac{1}{m} \sum_{q=1}^m (\nabla_\gamma P(\gamma(\alpha)))_{\gamma(\alpha)=(1-\alpha)x+\alpha\bar{x}, \alpha=q/m} \right]_i [\bar{x} - x]_i \right| \tag{12}$$

$$= \sum_i^L g(x_i) \tag{13}$$

## A.2 PROOF OF THEOREM 2

**Proof**: The derivative $\partial \mathcal{L}_{click}/\partial l_i$ of the loss function with respect to the LLM click score $l_i$ can be calculated as:

$$\frac{\partial \mathcal{L}_{click}}{\partial l_i} = -\sum_{j=1}^2 \frac{\partial y_j \log(p_j)}{\partial l_i} = -\sum_{j=1}^2 y_j \frac{\partial \log(p_j)}{\partial l_i} = -\sum_{j=1}^2 y_j \frac{1}{p_j} \frac{\partial p_j}{\partial l_i}$$

$$= -\frac{y_i}{p_i} \frac{\partial p_i}{\partial l_i} - \sum_{j\neq i}^2 \frac{y_j}{p_j} \frac{\partial p_j}{\partial l_i} = -T \frac{y_i}{p_i} p_i(1 - p_i) - T \sum_{j\neq i}^2 \frac{y_j}{p_j}(-p_j p_i) \tag{14}$$

$$= -Ty_i + Ty_i p_i + T \sum_{j\neq i}^2 y_j p_i = -Ty_i + T \sum_{j=1}^2 y_j p_i = -Ty_i + Tp_i \sum_{j=1}^2 y_j$$

$$= T(p_i - y_i) \propto T$$

## A.3 PROOF OF THEOREM 3

**Proof**: We apply the SVD decomposition to the matrix $\mathbf{E} = \mathbf{U\Sigma V}^\top$. The orthogonal matrices $\mathbf{U}, \mathbf{V}^\top$ represents the rotation and reflections of the matrix $\mathbf{E}$ while the singular values $\sigma_1, \sigma_2..., \sigma_d$ control the magnitude of orthogonal directions. And we use the eigenvalues $\lambda_1, \lambda_2..., \lambda_j$ of the covariance matrix $\mathbf{\Sigma}' = \mathbf{E} \left[ \frac{1}{|\mathbb{B}|} \sum_{j=1}^{|\mathbb{B}|} \mathbf{e}_j \mathbf{e}_j^\top \right] = \frac{1}{|\mathbb{B}|} \mathbf{E}\mathbf{E}^\top \in \mathbb{R}^{d\times d}$ to replace the singular values.

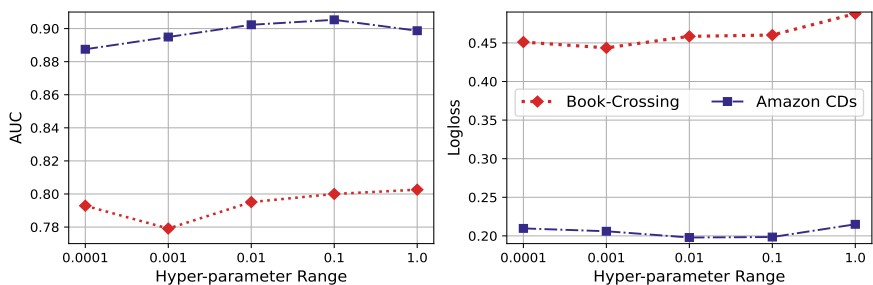

Figure 6: Hyper parameter sensitivity of SLLM4CTR with respect to $\beta$ on Book-Crossing and Amazon CDs dataset.

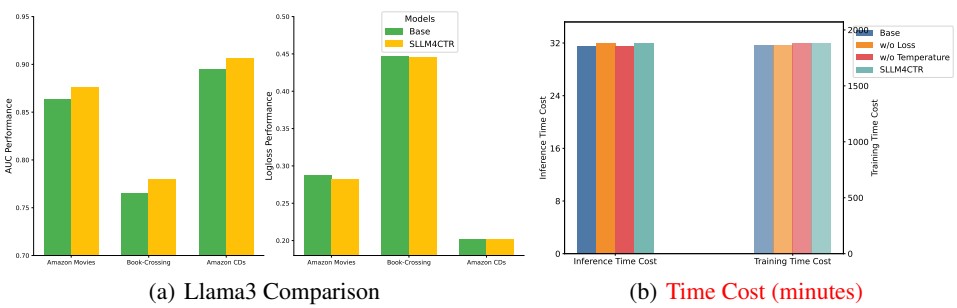

(a) Llama3 Comparison  (b) Time Cost (minutes)

Figure 7: Figure (a) shows the performance comparison about SLLM4CTR and its backbone on Llama3. Figure (b) shows the training and inference time cost about different variants of SLLM4CTR on 20-times expanded Book-Crossing dataset with batch size = 200.

$$Vol(\mathbf{E}) \propto \prod_{j=1}^{d} \sigma_j = \sqrt{\prod_{j=1}^{d} \lambda_j} = \sqrt{\det(\frac{1}{|\mathbb{B}|}\mathbf{E}\mathbf{E}^\top)} \tag{15}$$

Since there may be zero eigenvalues, we add the scaled identity matrix to slightly perturb the original matrix. But the added matrix also takes up space, we use the space volume of the added matrix as the unit volume to estimate the volume of $\mathbf{E}$. $\hat{\boldsymbol{\Sigma}} = \boldsymbol{\Sigma}' + \frac{\varepsilon^2}{d}\mathbf{I} = \frac{\varepsilon^2}{d}\mathbf{I} + \frac{1}{|\mathbb{B}|}\mathbf{E}\mathbf{E}^\top \in \mathbb{R}^{d \times d}$, such that the following equation holds:

$$Vol(\mathbf{E}) \propto \frac{Vol(\mathbf{E})}{Vol(Unit)} \propto \sqrt{\frac{\det\left(\frac{\varepsilon^2}{d}\mathbf{I} + \frac{1}{|\mathbb{B}|}\mathbf{E}\mathbf{E}^\top\right)}{\det\left(\frac{\varepsilon^2}{d}\mathbf{I}\right)}} = \sqrt{\det\left(\mathbf{I} + \frac{d}{|\mathbb{B}|\varepsilon^2}\mathbf{E}\mathbf{E}^\top\right)}. \tag{16}$$

### A.4 TEMPLATES

Book-Crossing: *Given the user's and item's attributes, identify whether the user will like the target item by answering Yes or No. Here is the information of the user [user_id]: The user is [age] years old and lives in [location]. Here is the information of the book [item_id]: The title is [title] written by [author] and published in [year of publication]. The publisher is [publisher].Response:[ ]*

Amazon CDs: *Given the user's and item's attributes, identify whether the user will like the target item by answering Yes or No. Here is the information of the user [user_id]. Here is the information of the item [item_id]: Its title is [title], and its price is [price]. Its description is [description]. Its sales rank in item is [sales rank].Response:[ ]*

### A.5 BASELINES

We incorporate the following representative baselines into comparison.

Table 4: Detailed datasets statistics.

| Datasets | Amazon Movies | Amazon CDs | Book-Crossing |
|---|---|---|---|
| User Number | 52494 | 61930 | 15101 |
| Item Number | 10845 | 11356 | 10614 |
| Interaction Number | 532248 | 616237 | 139475 |

Table 5: Additional results on three datasets.

| Model | Amazon Movies | | Book-Crossing | | Amazon CDs | |
|---|---|---|---|---|---|---|
| | AUC | Logloss | AUC | Logloss | AUC | Logloss |
| BAIR | 0.8458 | 0.2980 | 0.7409 | 0.4582 | 0.8835 | 0.2119 |
| KAC | 0.8592 | 0.2939 | 0.7545 | 0.4471 | 0.8904 | 0.2095 |
| FLIP | 0.8614 | 0.2877 | 0.7586 | 0.4483 | 0.8912 | 0.2064 |
| DIN | 0.8587 | 0.2922 | 0.7559 | 0.4505 | 0.8894 | 0.2126 |
| DIEN | 0.8614 | 0.2897 | 0.7587 | 0.4489 | 0.8920 | 0.2091 |
| w/ Contrastive Learning | 0.8598 | 0.2840 | 0.7668 | 0.4443 | 0.8904 | 0.2017 |
| w/o Label Matching Loss-1 | 0.8379 | 0.2638 | 0.7387 | 0.4595 | 0.8049 | 0.3595 |
| w/o Label Matching Loss-2 | 0.5236 | 0.3202 | 0.7551 | 0.4569 | 0.5268 | 0.4163 |

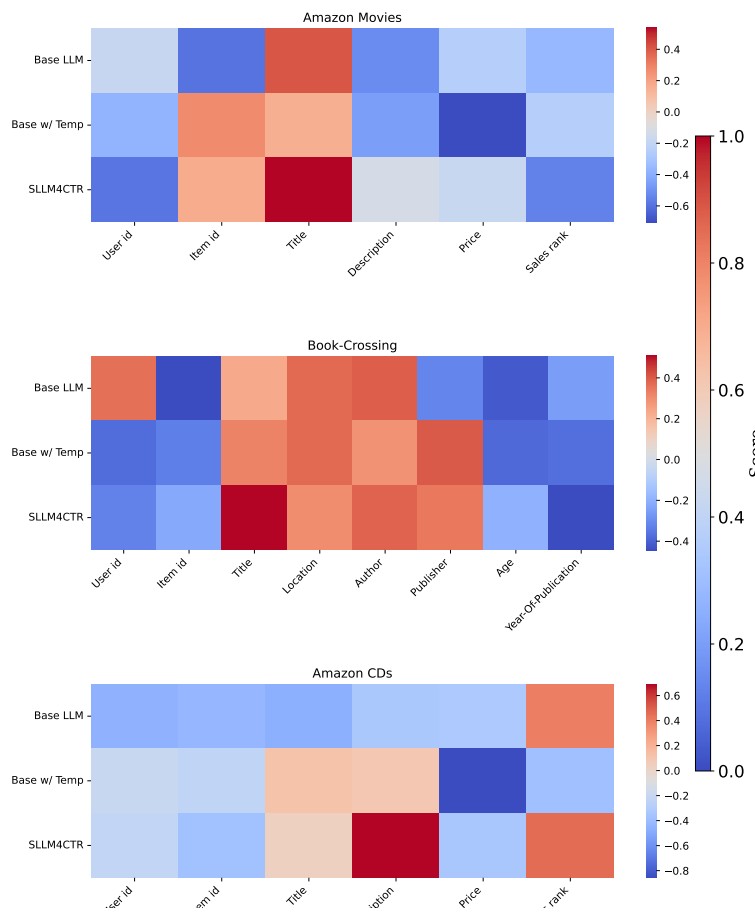

Figure 8: Comparison of feature attribution scores across base LLM, base LLM with temperature, and SLLM4CTR.

- **DeepFM** (Guo et al., 2017) contains the factorization machine layer (Juan et al., 2016) to learn the linear interaction and hidden layers to learn the non-linear interactions.

- **DCN** (Wang et al., 2021) incorporates the low-rank learnable matrix to the cross network and stacks the deep part on the wide part.

- **GDCNS** (Wang et al., 2023b) manages to effectively filter unimportant high-order feature interactions by incorporating the gated network.

- **GDCNP** (Wang et al., 2023b) contains a similar framework as the GDCNS and stacks the wide and deep parts in a parallel way.

- **FinalMLP** is an enhanced two-stream MLP model incorporating feature gating and interaction aggregation layers (Mao et al., 2023).

- **BAIU** leverages the language model to extract both open-box and black-box features from texts. The pre-train language model extracts the semantic features Yang et al. (2023a).

- **BAIR** employs contrastive learning to fine-tune the LLMs and align the review and item meta information (Bao et al., 2023b).

- **TallRec** employs the parameter-efficient fine-tuning techniques to fine-tune LLMs with an instruction tuning process (Bao et al., 2023b).

- **P5** designs multiple prompt format to adapt the variety of recommendation tasks on language models (Geng et al., 2022).

- **LlamaRec** first trains traditional CTR baseline and incorporates the result generated from baseline into the prompt (Yue et al., 2023).

- **ClickPrompt** leverages traditional CTR models as soft prompt generator and jointly training it with LLMs (Lin et al., 2024b).

- **KAC** uses hybrid-expert adaptor to transform LLM knowledge into a format compatible with traditional CTR. (Xi et al., 2024).

- **FLIP** combines both traditional CTR models and PLMs through feature-level alignment between ID-based models and PLMs. (Wang et al., 2024a).

- **DIN** introduces an attention mechanism to adaptively learn user interests from historical behaviors with respect to the target item. Zhou et al. (2018)

- **DIEN** improves upon DIN by modeling the temporal evolution of user interests through GRU with attentional update gates. Zhou et al. (2019)

- **w/o Label Matching Loss** adds the first design adaptive temperature to the base model.

- **w/o Adaptive Temperature** adds the second design label matching loss to the base model.

- **w/o Label Matching Loss-1** removes the first term in the label matching loss of SLLM4CTR.

- **w/o Label Matching Loss-2** removes the second term in the label matching loss of SLLM4CTR.

## A.6 IMPLEMENTATION DETAILS

We run all the models on NVIDIA Tesla A100 with 80G memory. For LLM-based CTR predictors, the Lora dropout rate, rank, and $\alpha$ are set as 0.2, 8, and 16 respectively. The weight decay is set as 1e-2. We implement the proposed framework in the pytorch and set the learning rate as 1e-4. The batch size is set as 32 and the optimizer is AdamW. We search the hyper-parameter $\alpha$, $\beta$ from the same range $\{1e-0, 1e-1, 1e-2, 1e-3, 1e-4\}$ by evaluating performance on the validation set. To control for the randomness, each model gets run ten times. And the reported results have passed the significance test with $p$-value $< 0.01$. For the traditional CTR-baselines, we search the hidden states of DNN from the range $\{512, 256, 128, 64, 32\}$, $L_2$ regularization term from the range $\{1e-0, 1e-1, 1e-2, 1e-3, 1e-4, 1e-4\}$ and the dropout ratio from the range $\{0, 0.05, 0.1, 0.15, 0.2, 0.25\}$. The batch size is set as 1024. The number of steps $m$ in the feature attribution score analysis section is set as 50. Because LLMs require the entire prompt as input, during the 50 steps, for the $q$-th step, we input $\frac{q}{50}$ times each input token embedding and compute the gradients of each token in the prompt for the $q$-th step. Then, the average of the gradients across these 50 steps is the token attribution score

## A.7 CASE STUDY

**Prompt Selection.** We provide two prompts: the first presents the unclicked sample with the highest predicted click probability, and the second presents the clicked sample with the lowest predicted click probability for the user in the first sample. Interestingly, the two items in these samples are nearly identical apart from their ID and title, yet their click prediction probabilities differ significantly.

**Prompt with Low Temperature.**

> Below is an instruction that describes a task, paired with an input that provides further context. Write a response that appropriately completes the request.
>
> Instruction: Given the user's and item's attributes, identify whether the user will like the target item by answering "Yes." or "No."
>
> Input: Here is the information of the user 126: The user is 39 years old and lives in Burlington, Ontario, Canada. Here is the information of the book 107: The title is Full House (Janet Evanovich's Full Series) written by Janet Evanovich, published in 2002. The publisher is St. Martin's Paperbacks. Response: No
>
> Original click prediction probability: 0.7519; Temperature: 0.7833

**Prompt with High Temperature.**

> Below is an instruction that describes a task, paired with an input that provides further context. Write a response that appropriately completes the request.
>
> Instruction: Given the user's and item's attributes, identify whether the user will like the target item by answering "Yes." or "No."
>
> Input: Here is the information of the user 126: The user is 39 years old and lives in Burlington, Ontario, Canada. Here is the information of the book 681: The title is Hot Six: A Stephanie Plum Novel written by Janet Evanovich, published in 2001. The publisher is St. Martin's Paperbacks. Response: No
>
> Original click prediction probability: 0.2480; Temperature: 10.2084

**Temperature Calibration.** We infer that LLMs may rely on limited features in CTR predictions for the first prompt, leading to significant differences between the predicted click probabilities and the labels, possibly due to inadequate feature modeling. To address this, we apply feature modeling to derive a relatively low temperature, which is then used to calibrate the predictions, smoothing the predicted click probabilities for this non-clicked sample and resulting in improved performance. For the second prompt, the features are effectively modeled, leading to a smaller difference between the predicted probability and the label. A larger temperature is assigned, further reducing the click probability of the sample with the non-click label, which also improves performance.

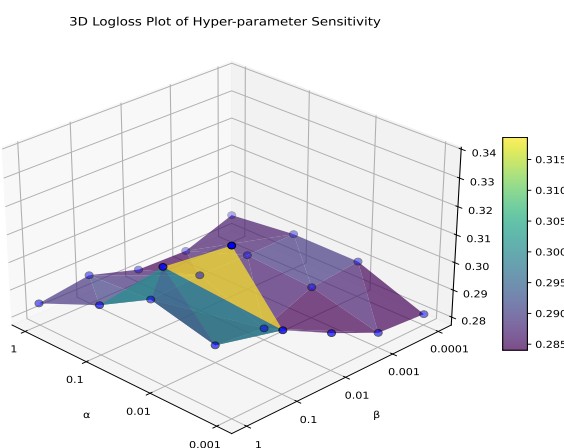

Figure 9: Figure shows hyper parameter sensitivity of SLLM4CTR w.r.t $\alpha$, $\beta$ on Amazon Movies dataset, presented as a three-dimensional plot using the Logloss metric. We have omitted the performance value when $\alpha = 0.0001$ since the model gets crashed in training.

## A.8 RELATED WORK

**LLMs for Recommendation.** Currently, there is a line of research focus that attempts to leverage the in-context learning ability of LLMs to enhance the performance (Wei et al., 2022; Lin et al., 2023). They aim to leverage retrieval-based methods by learning to access and utilize historical user-item interaction records (Salemi et al., 2024; Wu et al., 2024). Moreover, there is an emerging effort to fine-tune LLMs for different recommendation tasks (Yang et al., 2024), e.g., the job recommendation (Wu et al., 2023a), and the explainable recommendation (Li et al., 2023). Transferring LLM semantic knowledge towards small language models to improve their performance is another popular topic explored not only in collaborative filtering (Zheng et al., 2023; Zhu et al., 2024; Ren et al., 2024) but also in sequential recommendation (Lin et al., 2024c; Bao et al., 2023a; Zheng et al., 2024) or session-based recommendation (Sun et al., 2024). Recently, the application of LLMs as agents has gained significant attention. These approaches leverage large language models as planners to decide on and execute subsequent steps (Shi et al., 2024; Wang et al., 2024b). Our work instead focuses on fine-tuning LLMs for CTR prediction, a task with learning complex feature-click and feature-feature correlation. We demonstrate that fine-tuned LLM equipped with the proposed two simple designs can outperform traditional and LLM-based CTR predictors.

