# OpenReview forum: "Self-Monitoring Large Language Models for Click-Through Rate Prediction"
_ICLR.cc/2025/Conference — ICLR 2025 Conference Withdrawn Submission_

### Official Review · Reviewer_BVkm · 2024-10-28

**Soundness:** 2
**Presentation:** 3
**Contribution:** 2
**Rating:** 6
**Confidence:** 3

**Summary:**

The paper studies how to fine-tune LLM for CTR prediction task. The authors analyze the classical fine-tune approach can not predict well for tailed items, and introduce a self-monitoring approaches for improving fine-tune from two directions, one is adaptive temperature, the other is label matching loss. The experimental results on 3 public datasets demonstrate the advantages of the proposed SLLM4CTR method.

**Strengths:**

1. Important research question: how to fine-tune LLM for CTR prediction is practical for industry.
2. The analysis on comparation between simply  fine-tuned LLM for CTR and classical CTR models and feature attribution score are a good pilot studies and motivate the proposed methods.
3. The proposed methods (adaptive temperature and label match loss ) are easy to understand.
4. The writing is clear.

**Weaknesses:**

1. For the adaptive temperature T, it is better to give some examples, which prompt embedding has a higher T value, which has a lower T value? If the T value is extremely small, lyes/T is a extremly large, is there any threshold method for detailed implementation.
2. For hyperparameters analysis in Fig. 5, it is not clear the values of α, β and their corresponding AUC and logloss, it is better to use table or draw a 3D plot.

**Questions:**

As shown in the questions in weaknesses:
1. The authors could give some examples about prompts whose embedding has a higer or lower adaptive temperature T.
2. If T value is extremly large or small, do you need to calibrate it?
3. Fig. 5 is not very clear,  α, β in fomular 4 and 5 jointly effects the AUC and logloss, please show the grid search results.

---

### Official Review · Reviewer_Dtyf · 2024-10-29

**Soundness:** 3
**Presentation:** 2
**Contribution:** 3
**Rating:** 5
**Confidence:** 4

**Summary:**

This paper studies how to utilize the features in fine-tuning large language models for CTR prediction. This paper first identified two issues to improve the performance of LLM's ability: feature-wise, use all relevant features; click-wise, match user-item features to labels. Then, probe experiments on these two issues will be given; this paper proposes a framework, SLLM4CTR, which introduces adaptive temperature and label matching loss to improve performance. Finally,  extensive experiments are given to verify the effectiveness of this work.

**Strengths:**

1. This paper provides a clear motivation and some experiments on why they improve the performance from two aspects.

2. The proposed method is simple and easy to follow.

3. Experiments are extensive and well-study the problem.

**Weaknesses:**

1. The probe method for feature attribution score analysis is too simple. This is the motivation of the whole paper, so more details should be given. Why can Equation 1 give the attribution score? It should also give some proof.

2. The paper should be carefully proofread. For example, in Equation 2, $e_{click}$ is inconsistent in the denominator. In Table 4, the user numbers are 1/1/3, and the item numbers are 5/5/5, respectively. The description should be more precise.

3. The experimental results must give evidence that the proposed two issues are really resolved in the proposed method.

**Questions:**

1. In Figure 1, the best traditional CTR baseline really outperforms fine-tuned LLM in both head and tail items. However, in Table 2 and Table 3, traditional CTR baselines almost cannot outperform LLM-based baselines. Why does this happen? In clickpromt, authors claimed that semantic information is helpful in tail item prediction compared with traditional CTR models.

2. The prompt strategy is really important to fine-tune LLM for CTR tasks and also influence Equation 2 $T$. Should this be considered when the author argues feature attribution is important?

3. The datasets in this paper are too small to verify the cost of SLLMCTR because the click embedding matrix $\mathbold{E}$ is heavily related to the batch size, which is set to a small number of 32.

4. Can authors give an in-depth analysis of why adaptive temperature and label matching loss works?

---

### Official Review · Reviewer_98HJ · 2024-10-31

**Soundness:** 2
**Presentation:** 2
**Contribution:** 2
**Rating:** 5
**Confidence:** 4

**Summary:**

This paper introduces SLLM4CTR, which aims at improving the performance of LLMs on CTR prediction tasks. The authors identify key challenges in using LLMs for CTR prediction---the sparsity of click labels and inconsistent feature attention. The paper demonstrates that SLLM4CTR outperforms both traditional and LLM-based models on three real-world datasets (Amazon Movies, Book-Crossing, and Amazon CDs), particularly for "tail" items with limited click labels.

**Strengths:**

- The introduction of adaptive temperature and label matching loss are simple yet effective enhancements to the fine-tuning and inference process of LLMs.
-  The authors conducted experiments on three datasets and compared their method to both traditional CTR models and state-of-the-art LLM-based models. The results were presented clearly, and performance gains were demonstrated convincingly across head and tail items.

**Weaknesses:**

- While the authors provide the feature attribution score analysis, the setting and the motivation of the analysis are a bit unclear and confusing. More positive feature interactions do not necessarily result in better performance in CTR prediction. Then forcing the LLM to attend to more feature interactions is not so well motivated.
- While the paper compares SLLM4CTR to traditional CTR models (e.g., DeepFM, DCN), the exploration of non-LLM deep learning models for CTR prediction is somewhat shallow. Models like DIN (Zhou et al., 2018) or DIEN (Zhou et al., 2019), which are strong baselines in CTR tasks, are not discussed or included in the experiments. Adding these comparisons would provide a fuller picture of SLLM4CTR's relative performance.
- Although the authors mention that SLLM4CTR performs better on tail items, the paper lacks an in-depth analysis of why this is the case. A more detailed analysis of the improvements on tail items (e.g., feature utilization, attention distribution) would strengthen the claim and help clarify the benefits of the proposed approach.

**Questions:**

- [Q1] Could you please provide more explanation on the feature attribution score analysis? Why more positive feature interactions (as in traditional CTR models) are more feasible than fewer ones (as in LLM-based models)?
- [Q2] The paper asserts that SLLM4CTR improves feature utilization, particularly through adaptive temperature. However, it would be helpful if the authors could provide more clarification on how exactly the feature distribution changes after applying adaptive temperature. Could you provide visualizations of attention weights before and after applying adaptive temperature to highlight this effect?
- [Q3] The introduction of the label matching loss is promising, but it would be valuable to include a more detailed justification for the specific choice of this loss function. How does the label matching loss compare to other regularization techniques (e.g., contrastive loss)? Additionally, it would be insightful to see how the representation space evolves with and without the label matching loss via t-SNE or PCA visualizations.

---

### Official Review · Reviewer_2i8D · 2024-11-01

**Soundness:** 2
**Presentation:** 3
**Contribution:** 1
**Rating:** 3
**Confidence:** 4

**Summary:**

This paper targets the single-epoch phenomenon in the CTR domain. The author identifies the overfitting as coming from the embedding layer. Specifically, a multi-epoch data augmentation method(MEDA) is proposed to study the disentangling of the dependency between embedding and the MLP layer. Both an incremental and non-incremental approach are proposed. The proposed method has proven effective with both online and offline experiments.

**Strengths:**

1. The presentation of this paper is easy-to-follow.

**Weaknesses:**

1. The major weaknesses of this paper are that the baselines are too weak and the comparison is not fair. Traditional CTR models cannot directly encode textual features, such as *description*, into this prediction process. In contrast, LLMs are naturally suitable for this info. Given that the input to traditional and LLM-based models are not the same, it's hard to tell whether the improvement is caused by the textual features or the introduction of LLM into CTR prediction. A fair comparison would be encoding those textual features into CTR models.
2. The paper does not elaborate on the efficiency cost of introducing LLM. LLM-based CTR models are known to be heavy and unfit for real-world applications.
3. The experimental setup of baselines is biased towards LLM-based CTR models. LLM-based models are not able to be trained in large batch sizes. 32 is too small for traditional CTR models. Also, the LR and L2 of traditional models are not tuned, undermining the effectiveness of the proposed methods.

**Questions:**

1. What is the necessity of introducing LLM into CTR prediction models?
2. Why numbers in Table 4 are so strange?

---

### Official Review · Reviewer_2Bcd · 2024-11-04

**Soundness:** 2
**Presentation:** 2
**Contribution:** 2
**Rating:** 5
**Confidence:** 3

**Summary:**

This paper highlights two shortcomings of fine-tuned large language models (LLMs) in click-through rate (CTR) prediction. First, fine-tuned LLMs exhibit low attention to user-item features, failing to fully leverage these features. Second, they do not effectively match features with click labels. To address these issues, this paper proposes an adaptive temperature mechanism that associates the LLM’s attention on user-item features with its predictions. Additionally, it introduces a label matching loss to make click and non-click labels more distinguishable in the representation space.

**Strengths:**

1.	This paper insightfully identifies two key limitations of fine-tuned LLMs in CTR prediction and proposes targeted solutions.
2.	The proposed SLLM4CTR improves the performance of LLMs in CTR prediction, showing significant gains over current baselines.

**Weaknesses:**

1. There is no analysis of training and inference costs. Large models have much higher inference overhead than traditional recommendation systems, making it impossible to  deploy SLLM4CTR online in CTR estimation scenarios with strict latency requirements, which limits the contribution of this work.
2. Some symbols or references in the paper lack clear explanations, increasing the difficulty for readers. (Q1)
3. The explanation of the underlying principles of the proposed adaptive temperature has some issues and requires a more reasonable clarification.
4. The relationship between the label matching loss and the problem it addresses—i.e., the insufficient matching of user-item features with click labels in LLMs—is unclear.

**Questions:**

1.	What does  x_q  represent in Equation (1)?  It would be better to provide a brief explanation of the principle behind integrated gradient attributions.
2.	Could the author explain why the adaptive temperature can increase the click prediction loss for low-confidence predictions (line 262)? For a low-confidence prediction with a click label of 1, assume the initial prediction distribution is  P_{\text{click}=1} = 0.6  and  P_{\text{click}=0} = 0.4 . A lower  T  would make the prediction distribution sharper, for example, changing it to  P_{\text{click}=1} = 0.9  and  P_{\text{click}=0} = 0.1 , which would reduce the loss—contradicting the author’s explanation.
3.	On line 269,  e_c  uses the embeddings of all tokens in the input rather than only the embeddings of the user-item feature tokens. This means that the cosine similarity measures attention to all input tokens, rather than specifically to the user-item features, which contradicts the author’s explanation.
4.	In Section 4.2, the author states that the purpose of the label matching loss is to make click and non-click labels easier to distinguish in the representation space. Could the author explain the relationship among distinguishing click and non-click labels in the representation space, the effectiveness of the LLM in matching user-item features with click labels, and the performance on tail items?

---

### Note · Authors · 2025-01-18

I have read and agree with the venue's withdrawal policy on behalf of myself and my co-authors.